# Bayesian Neural Networks with Variance Propagation for Uncertainty Evaluation

## Abstract

Uncertainty evaluation is a core technique when deep neural networks (DNNs) are used in real-world problems. In practical applications, we often encounter unexpected samples that have not seen in the training process. Not only achieving the high-prediction accuracy but also detecting uncertain data is significant for safety-critical systems. In statistics and machine learning, Bayesian inference has been exploited for uncertainty evaluation. The Bayesian neural networks (BNNs) have recently attracted considerable attention in this context, as the DNN trained using dropout is interpreted as a Bayesian method. Based on this interpretation, several methods to calculate the Bayes predictive distribution for DNNs have been developed. Though the Monte-Carlo method called MC dropout is a popular method for uncertainty evaluation, it requires a number of repeated feed-forward calculations of DNNs with randomly sampled weight parameters. To overcome the computational issue, we propose a sampling-free method to evaluate uncertainty. Our method converts a neural network trained using dropout to the corresponding Bayesian neural network with variance propagation. Our method is available not only to feed-forward NNs but also to recurrent NNs including LSTM. We report the computational efficiency and statistical reliability of our method in numerical experiments of language modeling using RNNs, and the out-of-distribution detection with DNNs.

## 1 Introduction

Uncertainty evaluation is a core technique in practical applications of deep neural networks (DNNs). As an example, let us consider the Cyber-Physical Systems (CPS) such as the automated driving system. In the past decade, machine learning methods are widely utilized to realize the environment perception and path-planing components in the CPS. In particular, the automated driving system has drawn a huge attention as a safety-critical and real-time CPS (NITRD CPS Senior Steering Group, 2012; Wing, 2009). In the automated driving system, the environment perception component is built using DNN-based predictive models.

In real-world applications, the CPS is required to deal with unexpected samples that have not seen in the training process. Therefore, not only achieving the high-prediction accuracy under the ideal environment but providing uncertainty evaluation for real-world data is significant for safety-critical systems (Henne et al., 2019). The CPS should prepare some options such as the rejection of the recommended action to promote the user's intervention when the uncertainty is high. Such an interactive system is necessary to build fail-safe systems (Varshney & Alemzadeh, 2017; Varshney, 2016).

On the other hand, the uncertainty evaluation is useful to enhance the efficiency of learning algorithms, i.e., samples with high uncertainty are thought to convey important information for training networks. Active data selection based on the uncertainty has been studied for long time under the name of active learning (David et al., 1996; Gal et al., 2017; Holub et al., 2008; Li & Guo, 2013; Shui et al., 2020).

In statistics and machine learning, Bayesian estimation has been commonly exploited for uncertainty evaluation (Bishop, 2006.). In the Bayesian framework, the prior knowledge is represented as the prior distribution of the statistical model. The prior distribution is updated to the posterior distribution based on observations. The epistemic model uncertainty is represented in the prior distribution,

and upon observing data, those beliefs can be updated in the form of a posterior distribution, which yields model uncertainty conditioned on observed data. The entropy or the variance is representative of uncertainty measures (Cover & Thomas, 2006). For complicated models such as DNNs, however, a direct application of Bayesian methods is prohibited as the computation including the high-dimensional integration highly costs.

In deep learning, Bayesian methods are related to stochastic learning algorithms. This relation is utilized to approximate the posterior over complex models. The stochastic method called *dropout* is a powerful regularization method for DNNs (Srivastava et al., 2014). In each layer of the DNN, some units are randomly dropped in the learning using stochastic gradient descent methods. Gal & Ghahramani (2016a) revealed that the dropout is interpreted as the variational Bayes method. Based on this interpretation, they proposed a simple sampling method of DNN parameters from the approximate posterior distribution. Furthermore, the uncertainty of the DNN-based prediction is evaluated using the Monte-Carlo (MC) method called *MC dropout*.

While the Bayesian DNN trained using dropout is realized by a simple procedure, the computational overhead is not ignorable. In the MC dropout, dropout is used also at the test time with a number of repeated feed-forward calculations to effectively sample from the approximate posterior. Hence, the naive MC dropout is not necessarily relevant to the system demanding the real-time response.

In this work, we propose a sampling-free method to evaluate the uncertainty of the DNN-based prediction. Our method is computationally inexpensive comparing to the MC dropout and provides reliable uncertainty evaluation. In the following, we will first outline related works. Section 3 is devoted to show the detailed formulae of calculating the uncertainty. In our method, an upper bound of the variance is propagated in each layer to evaluate the uncertainty of the output. We show that the our method alleviates the overconfident prediction. This property is shared with scaling methods for the calibration of the class-probability on test samples. In Section 4, we study the relation between our method and scaling methods. In Section 5, we demonstrate the computational efficiency and statistical reliability of our method through some numerical experiments using both DNNs and RNNs.

## 2 RELATED WORKS

The framework of Bayesian inference is often utilized to evaluate the uncertainty of DNN-based predictions. In Bayesian methods, the uncertainty is represented by the predictive distribution defined from the posterior distribution of the weight parameters. MacKay (1992) proposed a simple approximation method of the posterior distribution for neural networks, and demonstrated that the Bayesian method improves the prediction performance on classification tasks. Graves (2011) showed that the variational method efficiently works to approximate the posterior distribution of complex neural network models.

There are many approaches to evaluate the uncertainty of modern DNNs (Alex Kendall & Cipolla, 2017; Choi et al., 2018; Lu et al., 2017; Le et al., 2018). We briefly review MC-based methods and sampling-free methods.

**Monte-Carlo methods based on Stochastic Learning:** The randomness in the learning process can be interpreted as a prior distribution. In particular, the dropout is a landmark of stochastic regularization method to train DNNs (Srivastava et al., 2014). Gal & Ghahramani (2016a) proposed a simple method to generate weight parameters from the posterior distribution induced from the prior corresponding to the dropout regularization. The predictive distribution is approximated by the MC dropout, which compute the expected output over the Monte-Carlo sampling of the weight parameters. Gal & Ghahramani (2016b) reported that the MC dropout efficiently works not only for feed-forward DNNs but for recurrent neural networks (RNNs). Another sampling based method is the ensemble-based posteriors with different random seeds (Lakshminarayanan et al., 2017). However, the computation cost is high as the bootstrap method requires repeated training of parameters using resampling data.

**Sampling-free methods:** Though the MC dropout is a simple and practical method to evaluate the uncertainty, a number of feed-forward computations are necessary to approximate the predictive distribution. Recently, some sampling-free methods have been proposed for the uncertainty

evaluation. Probabilistic network is a direct way to deal with uncertainty. The parameters of the probabilistic model, say the mean and the variance of the Gaussian distribution, are propagated in probabilistic neural networks. Then, the uncertainty evaluation is given by a single feed-forward calculation. Choi et al. (2018) used the mixture of Gaussian distributions as a probabilistic neural network and Wang et al. (2016) proposed natural-parameter networks as a class of probabilistic neural networks based on exponential families. For a given input vector, the network outputs the parameters of the distribution. For the recurrent neural networks, Hwang et al. (2019) proposed a variant of the natural-parameter networks. Instead of parameters of statistical models, Wu et al. (2019) developed a sampling-free method to propagate the first and second order moments of the posterior distribution.

Sampling-free methods can evaluate the uncertainty with a one-pass computation for neural networks. However, specialized learning algorithms are required to train the probabilistic networks. Our method is applicable to DNNs and RNNs trained by common learning methods with the dropout. Postels et al. (2019) and Shekhovtsov & Flach (2019) proposed similar methods that propagate the uncertainty of the network to the output layer. Differently from the past works, our method takes the upper limit of the correlations among the inputs at the affine layer into account when the uncertainty is evaluated. In addition, we show that our method efficiently works even for RNNs.

## 3 UNCERTAINTY EVALUATION WITH VARIANCE PROPAGATION

In this work, we assume that we can access to the weight parameters in the DNN and the dropout probability in the training process. As the variance is a common measure of uncertainty, we propose a variance propagation algorithm for the trained DNN.

Implementation of our method called `nn2vpbnn` is presented in Section A in the appendix. In our method, we need only the DNN or RNN trained using dropout. Unlike various kinds of probabilistic NNs, we do not need any specialized training procedure to evaluate the uncertainty. This is a great advantage for our implementation. Furthermore, the representative values of the predictive distribution, i.e. the mean and variance, are obtained by a one-path feed-forward calculation. Hence, we can circumvent iterative Monte-Carlo calculations.

### 3.1 UNCERTAINTY IN AFFINE LAYER

Let us consider the output of the affine layer $\mathbf{y} = \boldsymbol{W}\mathbf{x} + \boldsymbol{b}$ for the random input $\mathbf{x}$, where $\boldsymbol{W} = (W_{ij}) \in \mathbb{R}^{\ell \times m}$ and $\boldsymbol{b} = (b_i)_{i=1}^{\ell} \in \mathbb{R}^{\ell}$. Suppose that the random vector $\mathbf{x}$ has the mean vector $\mathbb{E}[\mathbf{x}]$ and the variance covariance matrix $(\Sigma_{\mathbf{x}})_{i,j} = \mathrm{Cov}(\mathbf{x}_i, \mathbf{x}_j)$ for $i, j = 1, \ldots, m$. Then, the mean vector $\mathbb{E}[\mathbf{y}]$ and the variance covariance matrix $\Sigma_{\mathbf{y}}$ of $\mathbf{y}$ are given by $\mathbb{E}[\mathbf{y}] = \boldsymbol{W}\mathbb{E}[\mathbf{x}] + \boldsymbol{b}$ and $\Sigma_{\mathbf{y}} = \boldsymbol{W}\Sigma_{\mathbf{x}}\boldsymbol{W}^T$.

As the estimation of the full variable-covariance matrix is not necessarily reliable, we use only the variances of each $\mathbf{x}_i$ and an upper bound of the absolute correlation coefficient to evaluate the uncertainty. For $\boldsymbol{W} = (W_{ij})$, the variance $\mathrm{Var}[\mathbf{y}_i]$ is $\mathrm{Var}[\mathbf{y}_i] = \sum_j W_{ij}^2 \mathrm{Var}[\mathbf{x}_j] + \sum_{j,j':j \neq j'} W_{ij}W_{ij'}\mathrm{Cov}(\mathbf{x}_j, \mathbf{x}_{j'})$. Suppose the absolute correlation coefficient among $\mathbf{x}_1, \ldots, \mathbf{x}_m$ is bounded above by $\rho$, $0 \leq \rho \leq 1$. Using the relation between the correlation and variance, we have

$$\mathrm{Var}[\mathbf{y}_i] \leq \sum_j W_{ij}^2 \mathrm{Var}[\mathbf{x}_j] + \rho \sum_{j,j':j \neq j'} |W_{ij}||W_{ij'}|\sqrt{\mathrm{Var}(\mathbf{x}_j)}\sqrt{\mathrm{Var}(\mathbf{x}_{j'})}$$

$$= (1 - \rho)\sum_j |W_{ij}|^2 \mathrm{Var}[\mathbf{x}_j] + \rho\bigg(\sum_j |W_{ij}|\sqrt{\mathrm{Var}(\mathbf{x}_j)}\bigg)^2, \quad i = 1, \ldots, \ell. \quad (1)$$

Under the independent assumption, i.e., $\rho = 0$, the minimum upper bound is obtained. The prediction with a small variance leads to overconfident decision making. Hence, the upper bounding of the variance is important to build fail-safe systems. A simple method of estimating $\rho$ is presented in Section 3.5.

Using the above formula, the mean and an upper bound of the variance of $\mathbf{y}$ are computed using the mean and an upper bound of the variance of $\mathbf{x}$. In this paper, such a computation is referred to as

the *Variance Propagation* or VP for short. Let us define the variance vector of the $m$-dimensional random vector $x = (x_1, \ldots, x_m) \in \mathbb{R}^m$ by $\mathrm{Var}[x] = (\mathrm{Var}[x_1], \ldots, \mathrm{Var}[x_m]) \in \mathbb{R}^m$. Furthermore, we denote the concatenated vector of the mean and variance of $\mathbf{z}$ or its approximation as $\mathcal{U}(\mathbf{z})$, i.e., $\mathcal{U}(\mathbf{z}) = (\mathbb{E}[\mathbf{z}], \mathrm{Var}[\mathbf{z}])$. The VP at the affine layer is expressed by the function $\mathcal{T}_{\mathrm{aff}}$,

$$\mathcal{U}(\mathbf{y}) = (\boldsymbol{m}, \boldsymbol{v}) = \mathcal{T}_{\mathrm{aff}}(\mathcal{U}(\mathbf{x})), \tag{2}$$

where $\boldsymbol{m} = \boldsymbol{W}\mathbb{E}[x] + \boldsymbol{b} \in \mathbb{R}^m$ and each element of $\boldsymbol{v} \in \mathbb{R}^m$ is defined by equation 1.

The average pooling layer, global average pooling layer (Lin et al., 2013), and the batch normalization layer (Ioffe & Szegedy, 2015) are examples of the affine layer. Hence, the VP of the affine layer also works to evaluate the uncertainty of these layers.

The distribution of $y_i$ is well approximated by the univariate Gaussian distribution if the correlation among $\mathbf{x}$ is small (Wang & Manning, 2013; Wu et al., 2019). Based on this fact, the uncertainty of $y_i$ can be represented by the univariate Gaussian distribution $N(\mathbb{E}[y_i], \mathrm{Var}[y_i])$. In our method, the variance $\mathrm{Var}[y_i]$ of the approximate Gaussian is given by the variance $\boldsymbol{v}$ in equation 2.

## 3.2 Output of Dropout Layer

Let us consider the uncertainty induced from the dropout layer (Srivastava et al., 2014). The dropout probability is denoted by $p$. In the dropout layer, the $m$-dimensional random input vector $\mathbf{x} = (x_1, \ldots, x_m)$ is transformed by the element-wise product $\mathbf{z} = \mathbf{x}\mathbf{d}$, where $\mathbf{d} = (d_1, \ldots, d_m)$ is the i.i.d. Bernoulli random variables, i.e., $\mathrm{Bernoulli}(p)$. As $\mathbf{x}$ and $\mathbf{d}$ are independent, the VP in the dropout layer is given by $(\mathbb{E}[\mathbf{z}], \mathrm{Var}[\mathbf{z}]) = \mathcal{T}_{\mathrm{drop}}(\mathcal{U}(\mathbf{x}))$, where $\mathbb{E}[\mathbf{z}] = p\mathbb{E}[\mathbf{x}]$ and $\mathrm{Var}[\mathbf{z}] = p\mathrm{Var}[\mathbf{x}] + p(1-p)\mathbb{E}[\mathbf{x}]^2$.

According to the Bayesian interpretation of the dropout revealed by Gal et al. (2017), the approximate posterior distribution of the output from the affine layer trained using dropout is given by the distribution of the random variable $y_i = \sum_{j=1}^m W_{ij}x_j d_j + b_i$, $d_1, \ldots, d_m \sim \mathrm{Bernoulli}(p)$. The mean and the variance of $y_i$ satisfy $\mathbb{E}[\mathbf{y}] = p\boldsymbol{W}\mathbb{E}[\mathbf{x}] + \boldsymbol{b}$ and $\mathrm{Var}[y_i] \leq (1-\rho)\sum_j |W_{ij}|^2\mathrm{Var}[x_j d_j] + \rho(\sum_j |W_{ij}|\sqrt{\mathrm{Var}[x_j d_j]})^2$. Since the stochastic input and the weight parameter in the dropout layer are independent, one can exactly calculate the variance of the product using each expectation and variance. The VP at the affine layer with the dropout is given by the composite function,

$$(\boldsymbol{m}, \boldsymbol{v}) = \mathcal{T}_{\mathrm{aff}} \circ \mathcal{T}_{\mathrm{drop}}(\mathcal{U}(\mathbf{x})).$$

The uncertainty of $y_i$ is then represented by the Gaussian distribution $N(m_i, v_i)$. A similar formula is found in the uncertainty evaluation of the LSTM unit in Section 3.4 with the explicit expressions.

## 3.3 Uncertainty via Activation Functions

The nonlinear activation function is an important component of neural network models in order to achieve high representation ability and accurate prediction (Cybenko, 1989). The ReLU, sigmoid function, and their variants are common activation functions. In several works, the expectation and the variance of the output from activation functions have been calculated (Frey & Hinton, 1999; MacKay, 1992; Daunizeau, 2017). Let us introduce the transformed distribution by the ReLU and sigmoid function.

The ReLU function is defined by $\mathbf{y} = \max(\mathbf{x}, 0)$. For $x_i \sim N(\mathbb{E}[x_i], \mathrm{Var}[x_i])$, the exact expectation and variance of $\mathbf{y}$ are expressed by the probability density $\phi$ and the cumulative function $\Phi$ of the standard Gaussian distribution (Frey & Hinton, 1999; Wu et al., 2019): $\mathbb{E}[\mathbf{y}] = \mathbb{E}[x]\Phi(\mathbb{E}[x]/\sqrt{\mathrm{Var}[x]}) + \sqrt{\mathrm{Var}[x]}\phi(\mathbb{E}[x]/\sqrt{\mathrm{Var}[x]})$ and $\mathrm{Var}[\mathbf{y}] = (\mathbb{E}[x]^2 + \mathrm{Var}[x])\Phi(\mathbb{E}[x]/\sqrt{\mathrm{Var}[x]}) + \mathbb{E}[x]\sqrt{\mathrm{Var}[x]}\phi(\mathbb{E}[x]/\sqrt{\mathrm{Var}[x]}) - \mathbb{E}[\mathbf{y}]^2$ using the element-wise operations for the two vectors $\mathbb{E}[x]$ and $\mathrm{Var}[x]$.

The sigmoid function is defined by $y_i = s(x_i) = 1/(1 + e^{-x_i})$. For $x_i \sim N(\mathbb{E}[x_i], \mathrm{Var}[x_i])$, MacKay (1992) and Daunizeau (2017) derived the approximate expectation and variance of $\mathbf{y}$, $\mathbb{E}[\mathbf{y}] \approx s(\frac{\mathbb{E}[\mathbf{x}]}{\sqrt{1+c\mathrm{Var}[\mathbf{x}]}})$, $\mathrm{Var}[\mathbf{y}] \approx s(\frac{\mathbb{E}[\mathbf{x}]}{\sqrt{1+c\mathrm{Var}[\mathbf{x}]}})(1 - s(\frac{\mathbb{E}[\mathbf{x}]}{\sqrt{1+c\mathrm{Var}[\mathbf{x}]}}))(1 - \frac{1}{\sqrt{1+c\mathrm{Var}[\mathbf{x}]}})$, where the constant $c$ depends on the approximation method. The common choice is $c = \pi/8 \approx 0.393$, while

Daunizeau (2017) found $c = 0.368$ based on numerical optimization. In the same way, one can calculate approximate expectation and variance of $\tanh(\mathbf{y})$. The VP at the activation layer is expressed by $\mathcal{U}[\mathbf{y}] = \mathcal{T}_{\mathrm{act}}(\mathcal{U}[\mathbf{x}])$, where the operation $\mathcal{T}_{\mathrm{act}}$ depends on the activation function. The output $\mathcal{U}[\mathbf{y}]$ is defined by the above expectation and variance.

In the multiclass classification problems, the softmax function is commonly used at the last layer in DNNs. However, the expectation of the softmax function does not have analytic expression under the multivariate Gaussian distribution. Daunizeau (2017) utilized the approximate expectation of the sigmoid function to approximate the expected softmax output. However, the variance of the softmax function was not provided. In this paper, we interpret the multiclass classification problem as the multi-label problem and at the last layer, we use the sigmoid functions as many as the number of labels. Given the transformations $z_k \longmapsto s(z_k)$, $k = 1, \ldots, G$ at the last layer for the classification with $G$ labels, the prediction is given by the label that attains the maximum value of $s(z_k)$. The advantage of this replacement is that the reliable evaluation of the uncertainty is possible for the sigmoid function as shown above. In numerical experiments, we show that the multi-label formulation with several sigmoid functions provides a comparable prediction accuracy as the standard multi-class formulation using the softmax function, while it also gives a reliable uncertainty evaluation.

## 3.4 LSTM Unit with Dropout

The uncertainty evaluation of the Recurrent Neural Networks (RNNs) is an important task as the RNNs are widely used in real-world problems. This section is devoted to the uncertainty propagation in the LSTM unit when the dropout is used to train the weight parameters (Gal & Ghahramani, 2016b). According to Greff et al. (2017), the standard form of the LSTM unit is defined by

$$(\mathbf{i}\ \mathbf{f}\ \mathbf{g}\ \mathbf{o}) = (s\ s\ \tanh\ s) \circ (\mathbf{h}_{t-1}\quad \mathbf{x}_t) \begin{pmatrix} \widetilde{U}_{\mathbf{i}} & \widetilde{U}_{\mathbf{f}} & \widetilde{U}_{\mathbf{g}} & \widetilde{U}_{\mathbf{o}} \\ \widetilde{W}_{\mathbf{i}} & \widetilde{W}_{\mathbf{f}} & \widetilde{W}_{\mathbf{g}} & \widetilde{W}_{\mathbf{o}} \end{pmatrix},$$

$$\mathbf{h}_t = \mathbf{o}\tanh(\mathbf{c}_t), \quad \mathbf{c}_t = \mathbf{f}\mathbf{c}_{t-1} + \mathbf{i}\mathbf{g}$$

using the sigmoid function $s$, where the multiplication of two vectors is the element-wise operation and $\circ$ is the composition of the linear layer and the activation function, i.e., $\mathbf{i} = s(\mathbf{h}_{t-1}\widetilde{U}_{\mathbf{i}} + \mathbf{x}_t\widetilde{W}_{\mathbf{i}})$, $\mathbf{g} = \tanh(\mathbf{h}_{t-1}\widetilde{U}_{\mathbf{g}} + \mathbf{x}_t\widetilde{W}_{\mathbf{g}})$, etc. The matrices $\widetilde{W}$'s and $\widetilde{U}$'s are the input weights and recurrent weights, respectively. The vectors, $\mathbf{i}, \mathbf{f}, \mathbf{g}$, and $\mathbf{o}$, denote the input gate, forget gate, new candidate vector, and output gate. The cell state $\mathbf{c}_t$ and the hidden state $\mathbf{h}_t$ retain the long and short term memory.

Here, $\widetilde{U}_*$ and $\widetilde{W}_*$ are regarded as random matrices distributed from the posterior distribution induced from the dropout using $\mathrm{Bernoulli}(p)$. Hence, each row of $\widetilde{U}_*$ and $\widetilde{W}_*$ are set to the null row vector with probability $1 - p$. When the tied dropout is used for LSTM, the same rows of all $\widetilde{U}_*$ are randomly dropped and the same rule is applied to $\widetilde{W}_*$. On the other hand, in the untied dropout layer, the dropout is separately executed for each $\widetilde{U}_*$ and $\widetilde{W}_*$. Detail of the tied and untied dropout is found in Gal & Ghahramani (2016b).

Let us consider the map from $\mathcal{U}(\mathbf{h}_{t-1}, \mathbf{c}_{t-1})$ to $\mathcal{U}(\mathbf{h}_t, \mathbf{c}_t)$. The map depends on the data $\mathbf{x}_t$. Since the computation in the LSTM with the dropout is expressed as the composite function of the dropout layer, affine layer and the activation function, we have

$$\mathcal{U}(\mathbf{i}, \mathbf{f}, \mathbf{g}, \mathbf{o}) = \mathcal{T}_{\mathrm{act}} \circ \mathcal{T}_{\mathrm{aff}} \circ \mathcal{T}_{\mathrm{drop}}(\mathcal{U}(\mathbf{h}_{t-1}, \mathbf{x}_t)).$$

Hence, the mean and variance vectors of $\mathbf{h}_t$ and $\mathbf{c}_t$ are obtained from those of $\mathbf{i}, \mathbf{f}, \mathbf{g}, \mathbf{o}$ and $\mathbf{c}_{t-1}$. This computation is shown below. We need an appropriate assumption to calculate $\mathbb{E}[\mathbf{c}_t]$ and $\mathrm{Var}[\mathbf{c}_t]$ as we do not use the correlations. The simplest assumption is the independence of random vectors. When $\mathbf{f}, \mathbf{c}_{t-1}, \mathbf{i}$ and $\mathbf{g}$ are independent, we obtain

$$\mathbb{E}[\mathbf{c}_t] = \mathbb{E}[\mathbf{f}\mathbf{c}_{t-1}] + \mathbb{E}[\mathbf{i}\mathbf{g}] = \mathbb{E}[\mathbf{f}]\mathbb{E}[\mathbf{c}_{t-1}] + \mathbb{E}[\mathbf{i}]\mathbb{E}[\mathbf{g}], \tag{3}$$

$$\mathrm{Var}[\mathbf{c}_t] = \mathrm{Var}[\mathbf{f}]\mathrm{Var}[\mathbf{c}_{t-1}] + \mathrm{Var}[\mathbf{f}]\mathbb{E}[\mathbf{c}_{t-1}]^2 + \mathbb{E}[\mathbf{f}]^2\mathrm{Var}[\mathbf{c}_{t-1}]$$

$$+ \mathrm{Var}[\mathbf{i}]\mathrm{Var}[\mathbf{g}] + \mathrm{Var}[\mathbf{i}]\mathbb{E}[\mathbf{g}]^2 + \mathbb{E}[\mathbf{i}]^2\mathrm{Var}[\mathbf{g}]. \tag{4}$$

This is the VP for the cell state vector $\mathbf{c}_{t-1}$ in the LSTM. Likewise, the VP for $\mathbf{h}_t$ is obtained. The above update function to compute the uncertainty of $\mathbf{h}_t$ and $\mathbf{c}_t$ from $\mathbf{i}, \mathbf{f}, \mathbf{g}, \mathbf{o}$ and $\mathbf{c}_{t-1}$ is denoted by $\mathcal{T}_{\text{cell}}$. As a result, we have

$$\mathcal{U}(\mathbf{h}_t, \mathbf{c}_t) = \mathcal{T}_{\text{cell}}(\mathcal{U}(\mathbf{i}, \mathbf{f}, \mathbf{g}, \mathbf{o}), \mathcal{U}(\mathbf{c}_{t-1})) = \mathcal{T}_{\text{cell}}(\mathcal{T}_{\text{act}} \circ \mathcal{T}_{\text{aff}} \circ \mathcal{T}_{\text{drop}}(\mathcal{U}(\mathbf{h}_{t-1}, \mathbf{x}_t)), \mathcal{U}(\mathbf{c}_{t-1})).$$

This is the VP formula from $(\mathbf{c}_{t-1}, \mathbf{h}_{t-1})$ to $(\mathbf{c}_t, \mathbf{h}_t)$. Repeating the above computation with the observed sequence $\{\mathbf{x}_t\}_{t=1}^T$, one can evaluate the uncertainty of the cell state vectors and the outputs $\{\mathbf{y}_t\}_{t=1}^T$, where $\mathbf{y}_t = \mathbf{h}_t$, $t = 1, \ldots, T$.

Let us consider the validity of the above independence assumption. For given $\mathbf{h}_{t-1}$, the conditional independence of $\mathbf{i}, \mathbf{f}, \mathbf{g}, \mathbf{o}$ and $\mathbf{c}_{t-1}$ holds when the untied dropout is used to train the LSTM unit, i.e., the equality $p(\mathbf{i}, \mathbf{f}, \mathbf{g}, \mathbf{o}, \mathbf{c}_{t-1}|\mathbf{h}_{t-1}) = p(\mathbf{c}_{t-1}|\mathbf{h}_{t-1}) \prod_{\mathbf{s} \in \{\mathbf{i}, \mathbf{f}, \mathbf{g}, \mathbf{o}\}} p(\mathbf{s}|\mathbf{h}_{t-1})$ holds for the posterior distribution. The randomness comes from the Bayesian interpretation of the untied dropout. Here, the observation $\mathbf{x}_t$ is regarded as a constant without uncertainty. Then, equation 3 and 4 exactly hold by replacing the mean and variance with conditional expectation and the conditional variance under the condition of $\mathbf{h}_{t-1}$. If the variance of $\mathbf{h}_{t-1}$ is small, the independence assumption is expected to be approximately valid. When the uncertainty of $\mathbf{h}_{t-1}$ is not ignorable, the sampling from the Gaussian distribution representing the uncertainty of $\mathbf{h}_{t-1}$ is available with the formulae $\mathbb{E}[\mathbf{c}_t] = \mathbb{E}_{\mathbf{h}_{t-1}}[\mathbb{E}[\mathbf{c}_t|\mathbf{h}_{t-1}]]$ and $\text{Var}[\mathbf{c}_t] = \mathbb{E}_{\mathbf{h}_{t-1}}[\text{Var}[\mathbf{c}_t|\mathbf{h}_{t-1}]]$ to compute $\mathbb{E}[\mathbf{c}_t]$ and $\text{Var}[\mathbf{c}_t]$ approximately.

### 3.5 Estimation of Correlation Parameter

When we evaluate the uncertainty in the affine layer, we need to determine the correlation parameter $\rho$ in equation 1. If the correlation of input to the affine layer is not ignored, the upper bound of the variance with an appropriate $\rho$ is used to avoid overconfidence. Hence, the estimation of the parameter $\rho$ is important. A simple method of estimating an appropriate $\rho$ is to use the validation set as follows.

1. For each candidate of $\rho$, execute the following steps.

   (a) For each data $(\boldsymbol{x}_i, \boldsymbol{y}_i)$ in the validation set, compute the mean vector $\boldsymbol{m}_i$ and variance vector $\boldsymbol{v}_i$ of the output of the network for given $\boldsymbol{x}_i$ using VPBNN.

   (b) Compute the predictive log-likelihood on the validation set,

   $$\text{LL}_\rho = \sum_{i:\text{validation set}} \log p(\boldsymbol{y}_i; \boldsymbol{m}_i, \boldsymbol{v}_i),$$

   where $p(\boldsymbol{y}; \boldsymbol{m}, \boldsymbol{v})$ is the probability density of the uncorrelated normal distribution with the mean $m_j$ and variance $v_j$ for each element.

2. Choose $\rho$ that maximizes $\text{LL}_\rho$.

Though we need to prepare several candidates of the correlation parameter, the computation cost is still lower than MC dropout. To evaluate the uncertainty on $N_{\text{test}}$ test samples, MC dropout with $T$ samplings requires $N_{\text{test}}T$ feed-forward calculations. The VPBNN with adaptive choice of $\rho$ needs approximately $2N_{\text{val}}K_{\text{cor}} + 2N_{\text{test}}$ feed-forward calculates, where $N_{\text{val}}$ is the number of the validation set and $K_{\text{cor}}$ is the number of candidates of $\rho$. The factor 2 comes from the computation of both mean and variance. Usually, $K_{\text{cor}}$ is much less than $T$ and $N_{\text{val}}$ is not extremely large in comparison to $N_{\text{test}}$. In practice, we do not need a large validation set to estimate $\rho$, as shown in numerical experiments. Hence, VPBNN with adaptive $\rho$ is computationally efficient than MC dropout. If distinct correlation parameters are used in each affine layer, the computation cost becomes large. In numerical experiments, we find that the uncertainty evaluation using the same $\rho$ in all the affine layers works well.

## 4 Scaling Methods for Calibration and Variance Propagation

The variance propagation is regarded as a calibration based on the uncertainty. There are some scaling methods for the calibration. Platt scaling (Platt, 1999) is a classical calibration method for multiclass classification problems, and the temperature scaling (Guo et al., 2017; Ji et al., 2019) is a simplified method of the Platt scaling.

Let us consider the conditional probability function $\Pr(y|\boldsymbol{z}) = e^{z_y}/\sum_{j=1}^{m} e^{z_j}$ for $\boldsymbol{z} = (z_1, \ldots, z_m) \in \mathbb{R}^m$ and $y = 1, \ldots, m$. The temperature scaling of $\Pr(y|\boldsymbol{z})$ is given by $\Pr(y|\boldsymbol{z}/T)$, where $T > 0$ is the temperature parameter. Usually $T$ is greater than one, and it softens the class probability. The Platt scaling of $\Pr(y|\boldsymbol{z})$ is defined by the scaling of $\boldsymbol{z}$ such that $\Pr(y|\boldsymbol{W}\boldsymbol{z} + \boldsymbol{b})$, where $\boldsymbol{W}$ is a diagonal matrix and $\boldsymbol{b}$ is a $m$-dimensional vector. The Platt scaling is the coordinate-wise calibration, while the temperature scaling is the homogeneous scaling for the feature vector $\boldsymbol{z}$. Another expansion of the temperature scaling is the bin-wise temperature scaling (Ji et al., 2019). In the bin-wise scaling, the sample space are divided into $K$ bins. The label probability is calibrated by $\Pr(y|\boldsymbol{z}/T_{k(\boldsymbol{z})})$ in which $k(\boldsymbol{z})$ is the index of the bin including the sample $\boldsymbol{z}$, and $T_k > 0$ is the temperature for the calibration at the $k$-th bin. Each $T_k$ is determined from validation data in the $k$-th bin. Intuitively, an extremely large label probability tends to yield an overconfident prediction. At the point $\boldsymbol{z}$ having the large maximum probability, the large scaling parameter $T_k$ is introduced to soften the overconfidence at the region including $\boldsymbol{z}$.

The calculation of the VP at the sigmoid activation layer for the multi-label classification is given by $s(\mathrm{z}_j) \longmapsto s(\mathbb{E}[\mathrm{z}_j]/\sqrt{1 + c\mathrm{Var}[\mathrm{z}_j]})$, $j = 1, \ldots, m$. When the uncertainty of the random vector $\boldsymbol{z}$ is not taken into account, the prediction using $s(\mathbb{E}[\mathrm{z}_j])$, $j = 1, \ldots, m$ is apt to be overconfident. Comparing to $s(\mathbb{E}[\mathrm{z}_j])$, the uncertainty defined by the variance $\mathrm{Var}[\mathrm{z}_j]$ works as a coordinate-wise calibration like the Platt scaling. If the variance is isotropic, the scaling does not change the ranking of the label probability like the temperature scaling.

In the variance propagation using the Taylor approximation (Postels et al., 2019), the class probability is calculated as the standard manner without calibration, while the variance is propagated along the layers in DNNs in order to evaluate the uncertainty. Hence, the calibration effect is not incorporated into the naive Taylor approximation method.

# 5 EXPERIMENTS

## 5.1 NUMERICAL EXPERIMENTS ON SYNTHETIC DATA

We assess the uncertainty computed by the VPBNN and the Taylor approximation using synthetic data. Let us consider the uncertainty evaluation for regression problems. The target function is the real-valued function on one-dimensional input space shown in Figure 1. We compared three methods; MC dropout with 1000 samples from the posterior distribution associated with the dropout, Taylor approximation, and VPBNN with adaptive $\rho$ and several fixed $\rho$'s. The architecture of the NN model is shown in Figure 4 of the Appendix. The results are presented in Figure 1. The Taylor approximation and the VPBNN with $\rho = 0$ tends to be overconfident and the VPBNN with $\rho = 0.15$ gives a similar result to the MC dropout. We find that the adaptive choice of $\rho$ can avoid the overconfidence, while providing a meaningful result.

In Section C in the appendix, we present the uncertainty evaluation for RNNs. Likewise, we find that the appropriate choice of $\rho$ relaxes the overconfident prediction and that the adaptive $\rho$ provides a meaningful result as well as MC dropout.

Overall, the VPBNN with appropriate $\rho$ provides similar results to the MC dropout. As the VPBNN needs only the one-path calculation as well as the VP using Taylor approximation, the computation cost is much less than the MC dropout. The adaptive choice of $\rho$ using the validation set efficiently works to produce a similar result as MC dropout.

Further numerical results are presented in Section B of the appendix. The Taylor approximation for the uncertainty evaluation proposed by Postels et al. (2019) also leads to a computationally efficient single-shot method to compute the uncertainty. However, we find that Taylor approximation tends to lead overconfident result compared to our method in the present experiments.

## 5.2 RNN FOR LANGUAGE MODELING

We report numerical experiments of language modeling problem. The problem setup is the same as the problem considered by Zaremba et al. (2014) and Gal & Ghahramani (2016b). We use Penn Treebank, which is a standard benchmark in this field. In the experiments, the LSTM consisting of two-layers with 650 units in each layer is used. The model architecture and most of hyper parameters

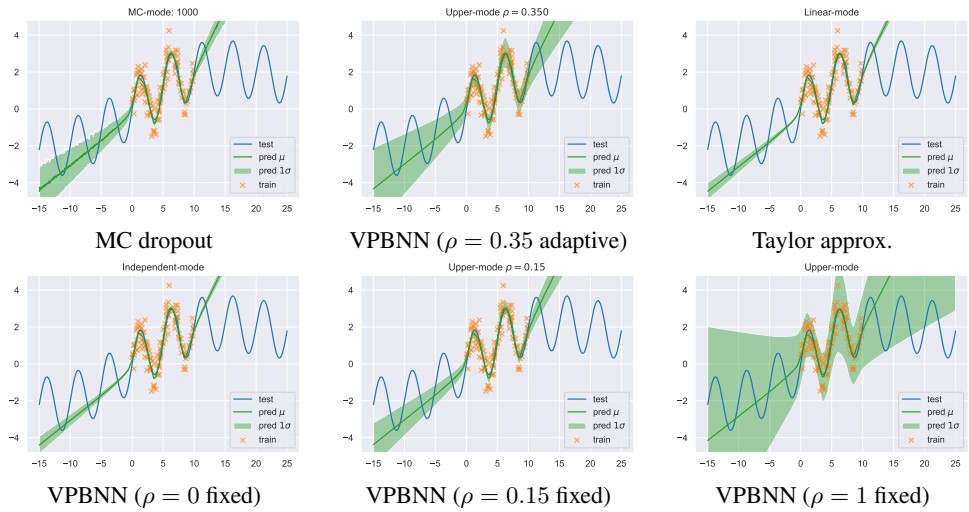

Figure 1: The uncertainty for feed-forward NNs is evaluated. The solid line is the target function and the training samples ($\times$) are plotted. For each method, the uncertainty is depicted as the confidence interval.

Table 1: Single model perplexity for the Penn Treebank language modeling task is presented. The asterisk ($*$) denotes the best perplexity on the test set for each dropout setting. The asterisk ($\circ$) means the perplexity reported in (Gal & Ghahramani, 2016b).

|  | untied weights | | tied weights | |
|---|---|---|---|---|
|  | Validation | Test | Validation | Test |
| MC dropout$^\circ$ | – | $78.6 \pm 0.1$ | – | $79.0 \pm 0.1$ |
| standard dropout approx.$^\circ$ | $81.9 \pm 0.2$ | $79.7 \pm 0.1$ | $81.8 \pm 0.2$ | $79.7 \pm 0.1$ |
| Taylor approx. | 82.65 | 79.34 | 82.63 | 79.67 |
| VPBNN: $\rho = 0$ | 81.30 | 78.05* | 81.09 | 78.20* |

are set to those used by Gal & Ghahramani (2016b). Figure 7 in the appendix shows the RNN and the converted VPBNN. The weight decay parameter is set to $10^{-7}$ according to the code in Github provided by the authors of Gal & Ghahramani (2016b), as the parameter was not explicitly written in their paper.

The results are shown in Table 1. The prediction performance is evaluated by the perplexity on the test set. In the table, the standard dropout approximation propagates the mean of each approximating distribution as input to the next layer (Gal & Ghahramani, 2016b). As the Taylor approximation computes the mean of the output without using the variance, it must provide the same result as the standard dropout approximation. In our experiment, both methods produced almost identical perplexity scores. This result means that we approximately reproduced the numerical results reported in the past papers. The MC dropout and the VPBNN with $\rho = 0$ achieved a lower perplexity than the others. Our method using only a one-path calculation can provide almost the same accuracy as the MC dropout that requires more than 1000 times feed-forward calculations of the output values.

Note that the VPBNN is not the approximation of MC dropout. Both MC dropout and VPBNN are an approximation of the posterior distribution, though MC dropout with a sufficient number of feed-forward calculations tends to provide a satisfactory result. The numerical experiments indicates that the number of feed-forward calculations in MC dropout is not sufficient for this task.

## 5.3 OUT-OF-DISTRIBUTION DETECTION

Let us consider the out-of-distribution detection problem. The task is to find samples whose distribution is different from that of the training samples. The uncertainty of samples is evaluated for this task. First of all, the neural network is trained using Fashion-MNIST dataset (Xiao et al., 2017). Then, several methods for uncertainty evaluation are used to detect samples from non-training

Table 2: Results of out-of-distribution detection are presented. "Test accuracy" is the prediction accuracy computed on the test set of Fashion-MNIST. For each pair of training domain (Fashion-MNIST) and non-training domain (MNIST, EMNIST, Kannada or Kuzushiji), the averaged AUC score computed using 30 random seeds is shown with the standard deviation. The asterisk $**$ (resp. $*$) denotes the highest (the second-highest) AUC for each non-training dataset.

| | Test accuracy | AUC score for each non-training domain | | | |
| | | MNIST | EMNIST | Kannada | Kuzushiji |
|---|---|---|---|---|---|
| MC100 | $0.923 \pm 0.002$ | $0.904 \pm 0.019$ | $0.928 \pm 0.012$ | $0.897 \pm 0.015$ | $0.965 \pm 0.006$ |
| MC2000 | $0.923 \pm 0.002$ | $0.916 \pm 0.022^*$ | $0.937 \pm 0.013^*$ | $0.909 \pm 0.015^*$ | $0.971 \pm 0.006^*$ |
| Taylor approx. | $0.923 \pm 0.002$ | $0.775 \pm 0.028$ | $0.833 \pm 0.017$ | $0.766 \pm 0.023$ | $0.860 \pm 0.017$ |
| VPBNN:adaptive | $0.923 \pm 0.002$ | $0.923 \pm 0.026^{**}$ | $0.946 \pm 0.016^{**}$ | $0.916 \pm 0.020^{**}$ | $0.981 \pm 0.005^{**}$ |

datasets. In this experiments, we use MNIST (Lecun et al., 1998), EMNIST-MNIST (Cohen et al., 2017), Kannada (Prabhu, 2019), and Kuzushiji (Clanuwat et al., 2018) as non-training datasets. The detection accuracy of each method is evaluated by the AUC measure on the test dataset.

We compared MC dropout with 100 sampling (MC100) or 2000 sampling (MC2000), Taylor approximation, and VPBNN with adaptive $\rho$. The network architecture is the CNN shown in Figure 8 of the Appendix. At the output layer, the multi-label sigmoid function is used. Numerical results for the softmax function are reported in Section E of the appendix. In the training process, Adam optimizer with early stopping on validation dataset is used. The 60k training data was divided into 50k training data for weight parameter tuning and 10k validation data for hyper-parameter tuning.

We confirmed that all methods achieve almost the same prediction accuracy on the test data of Fashion-MNIST. The result is shown in the "Test accuracy" column in Table 2. The prediction is done by using the top-ranked labels. Though MC dropout and VPBNN tend to relax the overconfident prediction, the calibration does not significantly affect the label prediction accuracy of this problem.

For the uncertainty evaluation, we used two criteria. One is the entropy computed from the mean value of the output, $\mathbb{H}[\mathbf{y}] = -\frac{1}{m}\sum_{i=1}^{m}\{\mathbb{E}[\mathbf{y}_i]\log\mathbb{E}[\mathbf{y}_i] + (1-\mathbb{E}[\mathbf{y}_i])\log(1-\mathbb{E}[\mathbf{y}_i])\}$, for the output $\mathbf{y} = (\mathbf{y}_i,\ldots,\mathbf{y}_m)$ of the NN, and the other is the mean-standard deviation (mean-std) (Kampffmeyer et al., 2016; Gal et al., 2017) that is the averaged standard deviation, i.e., $\sigma(\mathbf{y}) = \frac{1}{m}\sum_{i=1}^{m}\sqrt{\mathrm{Var}[\mathbf{y}_i]}$. In Section E of the appendix, we report the results of other uncertainty measure using not only sigmoid function but the softmax functions. Overall, we find that the mean-standard deviation outperforms the entropy measure in the out-of-distribution detection. This is because the uncertainty is rather related to the variance than expectation of the output value. Table 2 shows the results of the mean-standard deviation. In the adaptive VPBNN of this task, the estimated correlation parameter $\rho$ approximately ranges from 0.0 to 0.0005. Hence, VPBNN with the independent assumption, i.e., $\rho = 0$, also works well. Taylor approximation method fails to detect the sample from non-training distribution, This is because the estimation accuracy of the variance is not necessarily high as shown in Section B of the appendix.

Let us consider the computation cost. In our experiments, the computation time of MC dropout with 100 sampling is $15.6[\mathrm{sec}] \pm 28.3[\mathrm{ms}]$ on average for the uncertainty evaluation of 10K test samples. For MC dropout with 2000 sampling, the computation cost is approximately 20 times higher since it is proportional to the number of sampling. For the adaptive VPBNN, the computation time is $175[\mathrm{ms}] \pm 31.7[\mathrm{ms}]$ including the adaptive choice of $\rho$ from 10 candidates when 10K validation samples are used. VPBNN with adaptive $\rho$ provides comparable performance to MC dropout using a sufficient number of sampling while keeping much less computation cost. As discussed in language modeling, the number of feed-forward calculations in MC dropout is considered not sufficient for this task.

## 6 CONCLUSION

We developed a sampling-free method for uncertainty evaluation. Our method requires only a one-path calculation of DNNs or RNNs, while the MC-based methods need thousands of feed-forward calculations to evaluate the uncertainty. In the numerical experiments, we show that our method provides more reliable results than existing sampling-free methods such as the Taylor approximation.

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

## A    Implementation of NN2VPBNN

Using Keras (Chollet et al., 2015), we implemented nn2vpbnn that converts the DNN or RNN trained using dropout to the corresponding Bayesian neural network with variance propagation, i.e., VPBNN. Each layer of the VPBNN propagates the mean and the variance from the input layer to the output layer. The dropout layer in the VPBNN is the operation defined by $\mathcal{T}_{\mathrm{drop}}$. The output of VPBNN provides the mean and variance of the predictive distribution for the original BNN. In each layer, the VPBNN with zero variance provides the output without uncertainty. Figure 2 is an example of the VPBNN for the LSTM model. In the VPBNN, the input and output size is doubled after the dropout layer. This is because information on the uncertainty is added as the variance. Similarly, nn2vpbnn works to convert CNN models to the corresponding VPBNN.

At the affine layer, our method allows the users to tune the degree of the dependence among input variables by setting the parameter $\rho$. This parameter is determined according to the balance between the validity of the independence assumption and the safety required to the system. On the numerical experiments shown in Figure 6, setting $\rho = 0$ tends to produce an over-confident prediction, while the parameter setting with $\rho = 1$ corresponds to the least-confident prediction.

In the BNN, the dropout layer is the main source of the uncertainty. Besides the dropout layer, the batch normalization layer also has a Bayesian interpretation (Teye et al., 2018). Gaussian Dropout layer and Gaussian Noise layer also yield the uncertainty to the output of the layer, while these layers do not necessarily have the Bayesian interpretation. One can easily implement the variance propagation for these layers as a part of `nn2vpbnn`.

In our method, we utilize neural networks trained with dropout. Unlike various kinds of probabilistic NNs, we do not need any specialized training procedure to evaluate the uncertainty. This is a great advantage for our implementation. Furthermore, the representative values of the predictive distribution, i.e. the mean and variance, are obtained by the one-path feed-forward calculation. Hence, we can circumvent iterative Monte-Carlo calculations. These advantages are shared with the Taylor approximation method by Postels et al. (2019).

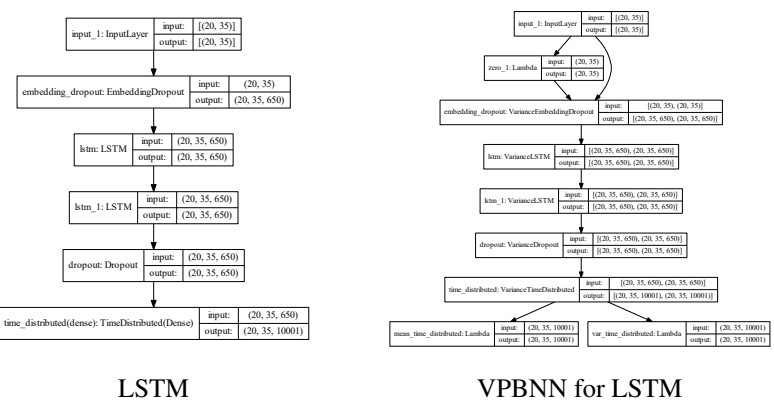

LSTM VPBNN for LSTM

Figure 2: Left: LSTM. Right: the corresponding VPBNN.

## B  SUPPLEMENTARY ON APPROXIMATION ACCURACY

Let us consider the numerical accuracy of VPBNN and Taylor approximation. The target is to compute the mean and variance of output value $\mathbf{y} = f(\boldsymbol{W}\mathbf{x} + \boldsymbol{b})$ for random input vector $\mathbf{x}$. The ReLU or sigmoid function is used as the activation function $f$. These two methods are compared with the Monte-Carlo method with sufficiently many samples. The distribution of input variable is the Gaussian distribution or the uniform distribution. The mean (resp. the standard deviation) of input variable varies in the interval $[-10, 10]$ (resp. $[0.1, 10]$) for both Gaussian distribution and the uniform distribution.

The absolute error of VPBNN and Taylor approximation from the MC method is shown in Figure 3. The horizontal axis and vertical axis denote the variance and mean of the input distribution, respectively. In numerical experiments, VPBNN achieved higher accuracy than Taylor approximation. We find that Taylor approximation tends to yield extremely large variance even when the sigmoid function is used as the activation function. Overall, VPBNN has a preferable property compared to the Taylor approximation.

The architectures of the NN and RNN used in Section 5.1 are shown in Figure 4 and 5.

## C  NUMERICAL EXPERIMENTS ON SYNTHETIC DATA: RNN

Numerical experiments of RNNs using synthetic data are also conducted. The RNN model is shown in Figure 5 of the Appendix. We evaluated the uncertainty of Bayesian RNN trained with dropout,

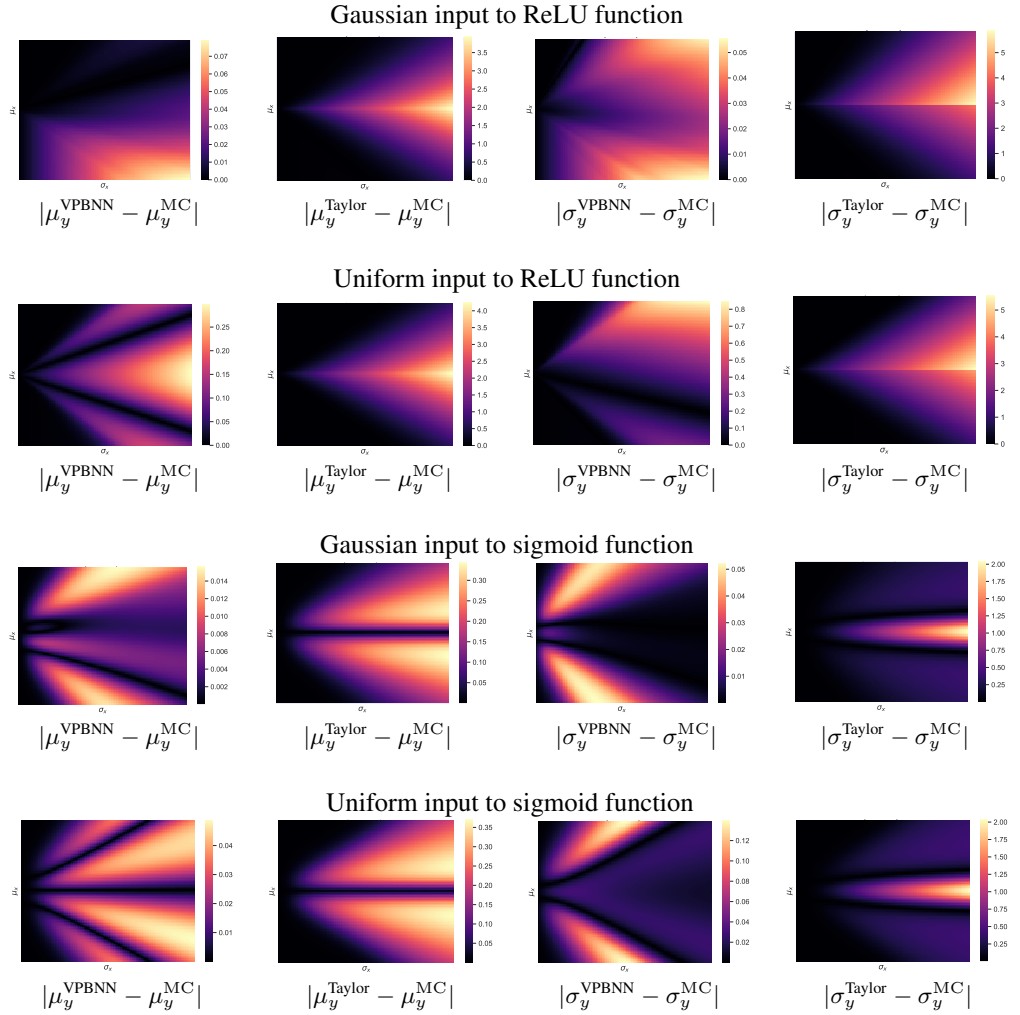

Figure 3: The mean and variance of the output from the ReLU or sigmoid function for the Gaussian or uniform inputs are computed using VPBNN and Taylor approximation. Absolute errors of VPBNN and Taylor approximation to the MC method are presented.

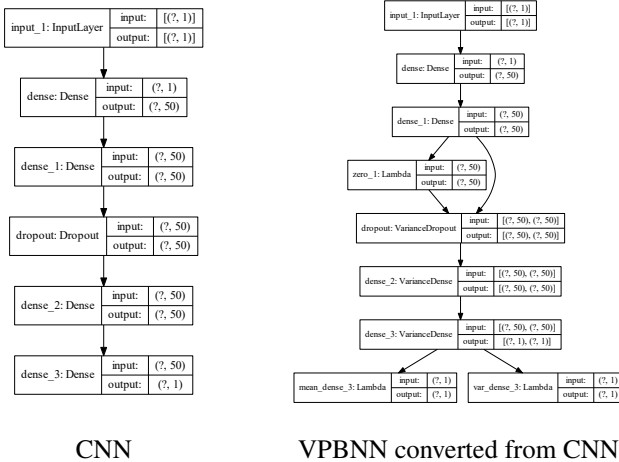

Figure 4: The architecture of NN to learn the regression function in Section 5.1. ReLU is used at the Dense of the middle layer and the identify function is used at the Dense in the last layer.

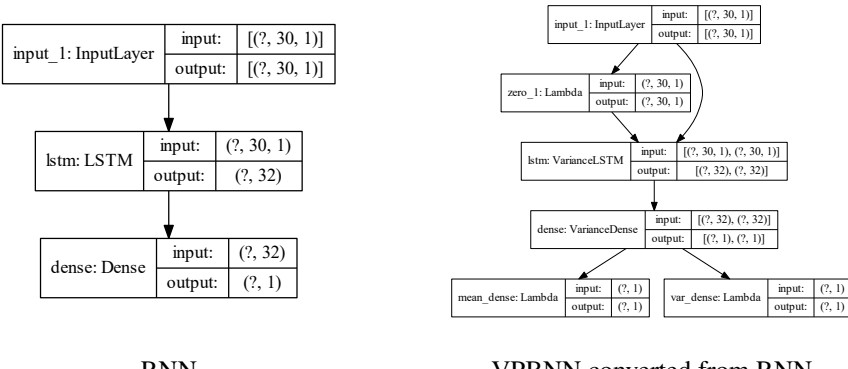

RNN                                    VPBNN converted from RNN

Figure 5: The architecture of RNN used in Section 5.1. The sigmoid function and $\tanh$ function are used as the activation function.

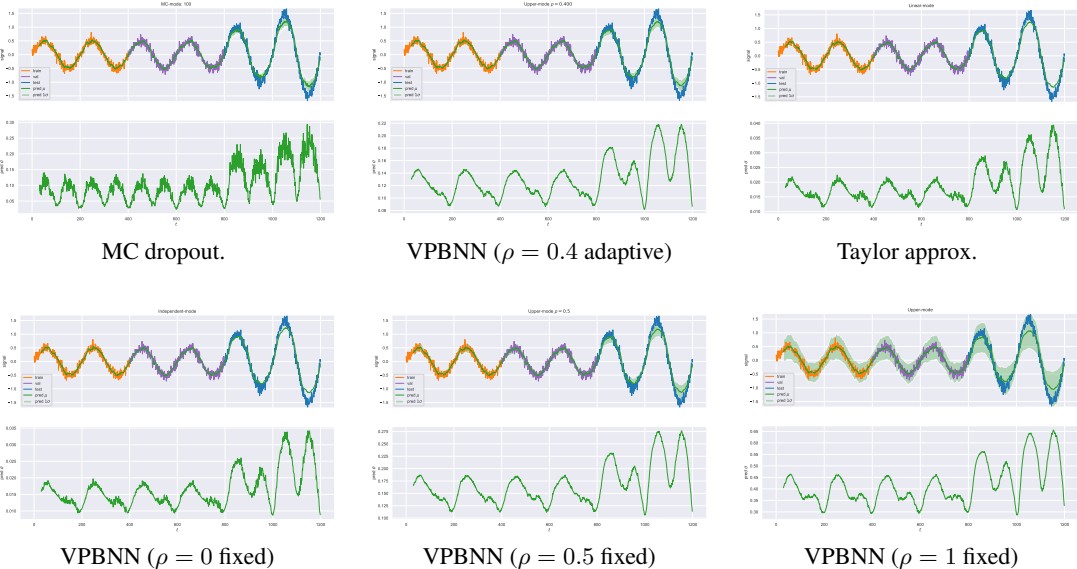

MC dropout.           VPBNN ($\rho = 0.4$ adaptive)         Taylor approx.

VPBNN ($\rho = 0$ fixed)        VPBNN ($\rho = 0.5$ fixed)        VPBNN ($\rho = 1$ fixed)

Figure 6: The uncertainty of Bayesian RNN is evaluated. The solid line is the target function. For each method, the uncertainty is depicted as the confidence interval. Upper panels: the target and estimated regression function with its uncertainty. Lower panels: the standard deviation of output value at each input value.

where the input is 30 length sequence. The results are shown in Figure 6. Again the Taylor approximation and the VPBNN with the independent assumption ($\rho = 0$) tend to yield overconfident results compared to the MC dropout. We find that the VPBNN with adaptive $\rho$ provides a similar results to the MC dropout.

## D    SUPPLEMENTARY OF RNN FOR LANGUAGE MODELING

The architectures of RNN used in Section 5.2 is shown in Figure 7.

## E    SUPPLEMENTARY OF OUT-OF-DISTRIBUTION

Let us consider the out-of-distribution detection. First of all, the neural network is trained using Fashion-MNIST (Xiao et al., 2017). Then, several methods for uncertainty evaluation are used to

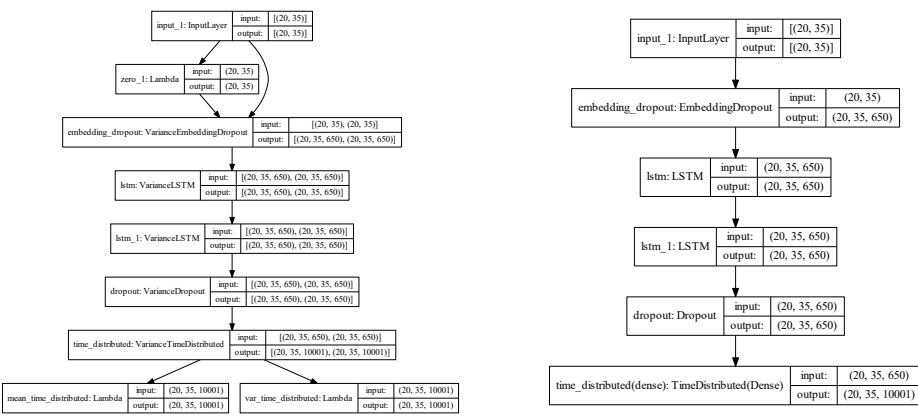

RNN used for Language Modeling       VPBNN from RNN

Figure 7: The architecture of RNN used for Language Modeling in Section 5.2. The sigmoid function and `tanh` function are used as the activation function.

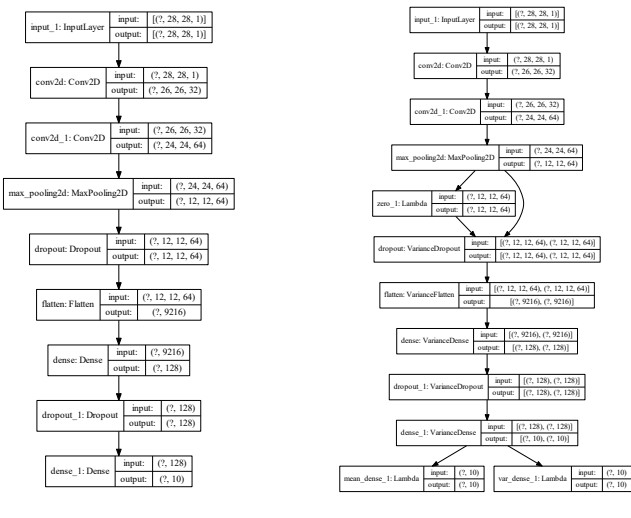

NN used for out-of-distribution       VPBNN

Figure 8: Neural networks to train Fashion MNIST in Section 5.3. ReLU is used in the middle dense layer, and the softmax or sigmoid function is used at the output layer.

detect samples from non-training datasets, MNIST (Lecun et al., 1998), EMNIST-MNIST (Cohen et al., 2017), Kannada (Prabhu, 2019), and Kuzushiji (Clanuwat et al., 2018). The detection accuracy of each method is evaluated by the AUC measure on the test dataset.

The network architecture used in Section 5.3 is the CNN in Figure 8 of the Appendix. In addition to the CNN with the softmax function provided in Keras, we implemented an another CNN with multi-label sigmoid functions at the output layer.

In the training process of the NNs, Adam optimizer with early stopping on validation dataset is exploited. The 60k training data was divided into 50k training data for the weight parameter tuning and 10k validation data for hyper-parameter tuning. For the uncertainty evaluation, we used two criteria; one is the entropy computed from the mean value of the output, and the other is the mean-standard deviation (mean-std) (Kampffmeyer et al., 2016; Gal et al., 2017) that is computed from

the variance. More precisely, the entropy defined from the softmax output is

$$\mathbb{H}[\mathbf{y}] = -\sum_{i=1}^{m} \mathbb{E}[\mathbf{y}_i] \log \mathbb{E}[\mathbf{y}_i],$$

and the entropy defined from the sigmoid function for multi-label setting is given by

$$\mathbb{H}[\mathbf{y}] = -\frac{1}{m} \sum_{i=1}^{m} \left\{ \mathbb{E}[\mathbf{y}_i] \log \mathbb{E}[\mathbf{y}_i] + (1 - \mathbb{E}[\mathbf{y}_i]) \log(1 - \mathbb{E}[\mathbf{y}_i]) \right\}.$$

The mean-std is defined by

$$\sigma(\mathbf{y}) = \frac{1}{m} \sum_{i=1}^{m} \sqrt{\mathrm{Var}[\mathbf{y}_i]}.$$

The results are presented in Table 3 in the appendix. For the out-of-distribution detection, we find that the mean-std based method outperforms the other methods using entropy criterion. Moreover, our method provides the uncertainty by only one-path feed-forward calculation, while the MC dropout needs more than hundreds of feed-forward calculations. The Taylor approximation fails to detect the sample from non-training distribution. This is because the approximation accuracy of the Taylor approximation is not necessarily high as shown in Section B.

On the other hand, all the methods considered here achieve almost the same prediction accuracy on the test data of Fashion MNIST as shown in Table 4. The prediction is done by using the top-ranked labels. In this experiment, the calibration of the label probability does not significantly affect the ranking of the label probability.

Table 3: AUC score of Out-of-Distribution detection for MC dropout using 2000 sampling, Taylor approximation, and VPBNN. The standard deviation of the AUC score is calculated using 30 different random seeds. The asterisk $**$ (resp. $*$) denotes the highest (the second-highest) AUC for each pair of training and non-training dataset.

| Training $\rightarrow$ non-training dataset | Method | Uncertainty | Activation func. | AUC |
|---|---|---|---|---|
| Fashion $\rightarrow$ MNIST | MC dropout | Entropy | softmax | $0.866 \pm 0.019$ |
| | | | sigmoid | $0.796 \pm 0.024$ |
| | | Mean-std | softmax | $0.934 \pm 0.014**$ |
| | | | sigmoid | $0.916 \pm 0.022$ |
| | Taylor approx. | Entropy | softmax | $0.718 \pm 0.026$ |
| | | | sigmoid | $0.682 \pm 0.028$ |
| | | Mean-std | softmax | $0.728 \pm 0.025$ |
| | | | sigmoid | $0.775 \pm 0.028$ |
| | VPBNN | Entropy | sigmoid | $0.806 \pm 0.025$ |
| | (adaptive $\rho$) | Mean-std | sigmoid | $0.923 \pm 0.026*$ |
| Fashion $\rightarrow$ EMNIST | MC dropout | Entropy | softmax | $0.893 \pm 0.013$ |
| | | | sigmoid | $0.846 \pm 0.016$ |
| | | Mean-std | softmax | $0.941 \pm 0.011*$ |
| | | | sigmoid | $0.937 \pm 0.013$ |
| | Taylor approx. | Entropy | softmax | $0.791 \pm 0.019$ |
| | | | sigmoid | $0.755 \pm 0.021$ |
| | | Mean-std | softmax | $0.777 \pm 0.017$ |
| | | | sigmoid | $0.833 \pm 0.017$ |
| | VPBNN | Entropy | sigmoid | $0.856 \pm 0.016$ |
| | (adaptive $\rho$) | Mean-std | sigmoid | $0.946 \pm 0.016**$ |
| Fashion $\rightarrow$ Kannada | MC dropout | Entropy | softmax | $0.867 \pm 0.017$ |
| | | | sigmoid | $0.785 \pm 0.020$ |
| | | Mean-std | softmax | $0.931 \pm 0.013**$ |
| | | | sigmoid | $0.909 \pm 0.015$ |
| | Taylor approx. | Entropy | softmax | $0.725 \pm 0.027$ |
| | | | sigmoid | $0.672 \pm 0.023$ |
| | | Mean-std | softmax | $0.738 \pm 0.024$ |
| | | | sigmoid | $0.766 \pm 0.023$ |
| | VPBNN | Entropy | sigmoid | $0.796 \pm 0.020$ |
| | (adaptive $\rho$) | Mean-std | sigmoid | $0.916 \pm 0.020*$ |
| Fashion $\rightarrow$ Kuzushiji | MC dropout | Entropy | softmax | $0.913 \pm 0.014$ |
| | | | sigmoid | $0.879 \pm 0.014$ |
| | | Mean-std | softmax | $0.964 \pm 0.008$ |
| | | | sigmoid | $0.971 \pm 0.006*$ |
| | Taylor approx. | Entropy | softmax | $0.800 \pm 0.025$ |
| | | | sigmoid | $0.766 \pm 0.017$ |
| | | Mean-std | softmax | $0.809 \pm 0.023$ |
| | | | sigmoid | $0.860 \pm 0.017$ |
| | VPBNN | Entropy | sigmoid | $0.893 \pm 0.013$ |
| | (adaptive $\rho$) | Mean-std | sigmoid | $0.981 \pm 0.005**$ |

Table 4: Test accuracy on the test set of Fashion-MNIST.

| Method | activation func. | Test accuracy |
|---|---|---|
| MC dropout | softmax | $0.923 \pm 0.003$ |
| | sigmoid | $0.923 \pm 0.002$ |
| Taylor approx. | softmax | $0.923 \pm 0.003$ |
| | sigmoid | $0.923 \pm 0.002$ |
| VPBNN (adaptive $\rho$) | sigmoid | $0.923 \pm 0.002$ |

