# OpenReview forum: "Bayesian Neural Networks with Variance Propagation for Uncertainty Evaluation"
_ICLR.cc/2021/Conference — Reject_

### Official Review · AnonReviewer2 · 2020-10-27
**Limited scope and novelty; broader implications unclear.**

**Rating:** 4
**Confidence:** 4

**Review:**

This paper proposes a variational-approximation of Neural Network uncertainty dependent on the 'bayesian interpretation' of dropout. Using this approach uncertainty measurements are provided using 'variance propagation.' Validation is provided on synthetic data, and MNIST/FMNIST.

Although a genuine effort is made towards deriving uncertainty through 'variance propagation,' I think the proposed approach is rather ad-hoc. The proposed approach relies on a 'bayesian interpretation,' of dropout, and is of limited applicability requiring hard coded rules for propagating through each layer. Currently, rules are derived for a small set of neural network layers.

The chief issue with this work is the 'generality' of the work. The promised approach here is applicable to specifically neural networks trained with SGD using dropout. Thus although the proposed approach claims to evaluate uncertainty, it is rather estimating the uncertainty introduced by dropout using a variational approximation, and assuming this is the same as uncertainty. I find this assertion unfair to other works which attempt to estimate various types of uncertainties (e.g. epistemic) in a principled manner.

The proposed approach depends on the hyperparameter \rho, which requires additional tuning to allow variance propagation through affine (e.g., fully connected) layers. This additional hyperparameter gives some doubt whether the proposed approach is something which can be easily integrated into existing models.

In Table 1, how do the authors address that their approach (which is meant to be an approximation to MC) appears to outperform MC? Shouldn't an approximation perform worse?

Although the proposed approach outperforms MC sampling in detecting out of distribution sampling, I think this benchmark is rather unfair. I'd like to see comparisons to other works which quantify uncertainty. The issue is that this only compares against model uncertainty assuming the 'bayesian interpretation' of MC is valid. I would strongly like to view comparisons to other works in quantifying network uncertainty.

On the whole, the lack of generality of the proposed approach, as well as the limited scope of 'variational approximation of model uncertainty assuming bayesian interpretation of dropout,' causes me to be highly skeptical of this work.

---

> ### Author Response · Authors · 2020-11-21
> **To reviewer 2**
>
> Thank you very much for your constructive feedback and comments.
>
> - The chief issue with this work is the 'generality' of the work. (omitted) I find this assertion unfair to other works which attempt to estimate various types of uncertainties (e.g. epistemic) in a principled manner.
>
> We provided the formula of uncertainty propagation for major layers used in DNNs. The SGD with mini-batch is quite common, and the dropout is a standard technique for DNN learning. Hence, our method has a wide range of applications.
> As the reviewer pointed out, there are many attempts to estimate various types of uncertainties (e.g., epistemic) in a principled manner.
> In our paper, we focused on the epistemic uncertainty of DNNs.
> For instance, probabilistic neural networks (PNN) are popular methods for assessing epistemic uncertainty. However,
> We need to prepare the specific network architecture and learning algorithm for PNN learning, while our approach exploits the standard network architectures and learning algorithms. In numerical experimenters, we compared our method with existing methods, which exploit the standard network architectures and learning algorithms such as Postels et al. (2019).
>
> - The proposed approach depends on the hyper-parameter rho.
>
> The implementation to incorporate the parameter rho is very simple. That is shown in Appendix A of our paper.  As for the estimation of \rho, there are several approaches to determine the relevant rho with a light computation. A simple method is to use the validation data to estimate \rho. In Section 3.5 of the revised paper, we explained an estimation method of \rho and added some numerical results in Section 5.
>
>
> - In Table 1, how do the authors address that their approach (which is meant to be an approximation to MC) appears to outperform MC?
>
> The VPBNN is not necessarily for the approximation of MC dropout. Both MC dropout and VPBNN approximate the true posterior distribution, though MC dropout with a sufficient number of feed-forward calculations tends to provide a satisfactory result. We supplemented this fact in the revision.
>
> - Although the proposed approach outperforms MC sampling in detecting out of distribution sampling, I think this benchmark is rather unfair (omitted) I'd like to see comparisons to other works which quantify uncertainty
> - I would strongly like to view comparisons to other works in quantifying network uncertainty.
>
> In our paper, our interest is to develop a method of assessing the uncertainty for the DNNs having the standard network architecture trained by the common methods such as SGD with dropout. Hence, we compared our method with the MC dropout proposed by Gal & Ghahramani (2016a) and the Tayler approximation by Postels et al. (2019).

---

### Official Review · AnonReviewer1 · 2020-10-28
**The paper doesn't differentiate itself well from existing work and misses key comparisons.**

**Rating:** 4
**Confidence:** 4

**Review:**

The paper proposes a sampling free approach for estimating predictive uncertainties in Bayesian neural networks trained via Monte-Carlo dropout. In particular,  given a dropout trained neural network, the paper develops a deterministic approximation to the test time predictive distribution that is otherwise approximated through Monte-Carlo simulations.

The paper is clearly written and proposes a solution to an important problem. Extracting fast and reliable predictive uncertainties from BNNs, trained via MC-dropout or other approximate inference techniques, is crucial for deploying BNNs in resource constrained and/or real time applications. Deterministic approximations like the one presented here are promising.

That being said, I have several concerns with the paper that revolve around differentiation from previous work,  missing empirical comparisons, and some curious modeling choices.

* The primary contribution of the paper appears to be the amortization of the MC-dropout posterior predictive density. This is achieved by propagating  uncertainty (first and second moments of the inputs) through the network.  The technical details of such uncertainty propagation have been worked out by several authors in the past (see 1, 2, 3, and the papers cited by the authors Wu et al., 2019 and Shekhovtsov & Flach 2019). While the application to amortizing MC-dropout is interesting, it appears to be a direct application of previous work.
* The notion of upper bounding the variance during the propagation is indeed distinct from previous work, but is neither principled nor empirically vetted to be consistently useful.  The experiments provide limited evidence in support of using the upper-bound. The derived uncertainty seems to be crucially dependent on the weighting factor $\rho$ and it is unclear how one would select this weighting factor. In general, injecting additional noise does not guarantee better calibrated uncertainties. Neither the language modeling task nor the OOD detection task make use of the upper bound.
* Amortization of the posterior predictive distribution is not a new idea. A popular approach is to use distillation [4, 5] based techniques to approximate the posterior predictive distribution with a second neural network. This work needs to be both discussed (advantages / disadvantages of the proposed approach over distillation)  and empirically compared against.
* Since the goal of the paper is to sufficiently well approximate the posterior predictive density, the experimental section would benefit from including calibration metrics (ECE / Brier score; see 6).
* (Minor) The paper at several places claims that the proposed approach “Unlike various kinds of probabilistic NNs, we do not need any specialized training procedure to evaluate the uncertainty”. While this is technically true, this is also true for other distillation based amortization techniques. Which arguably are simpler because one can just use an off the shelf network and not have to worry about propagating moments correctly.
* (Minor) Since the categorical Gaussian integral is intractable, the authors replace the softmax layer with a series of independent sigmoids and use an approximation for the sigmoid Gaussian convolution. This is strange. If we are going down this path, why not use independent probits instead of sigmoids? The probit-Gaussian integral is exactly computable and requires no further approximations (see equation 3.82 in http://www.gaussianprocess.org/gpml/chapters/RW3.pdf).

[1] https://papers.nips.cc/paper/5269-expectation-backpropagation-parameter-free-training-of-multilayer-neural-networks-with-continuous-or-discrete-weights.pdf

[2] https://arxiv.org/abs/1502.05336

[3] https://www.aaai.org/ocs/index.php/AAAI/AAAI16/paper/view/12391

[4] https://arxiv.org/abs/1506.04416

[5] https://arxiv.org/abs/2005.08110

[6] Tilmann Gneiting, Fadoua Balabdaoui, and Adrian E Raftery. Probabilistic forecasts, calibration and sharpness. Journal of the Royal Statistical Society: Series B (Statistical Methodology), 69(2):243–268, 2007.

---

> ### Author Response · Authors · 2020-11-21
> **To reviewer 1**
>
> Thank you very much for your constructive feedback and comments.
>
> - The technical details of such uncertainty propagation have been worked out by several authors in the past (see 1, 2, 3, and the papers cited by the authors Wu et al., 2019 and Shekhovtsov & Flach 2019).
> - While the application to amortizing MC-dropout is interesting, it appears to be a direct application of previous work.
> - The notion of upper bounding the variance during the propagation is indeed distinct from previous work, but is neither principled nor empirically vetted to be consistently useful.
> - Amortization of the posterior predictive distribution is not a new idea.
>
> We introduced the correlation parameter \rho to avoid the overconfident prediction. There are several approaches to determine the relevant rho with a light computation. A simple method is to use the validation data to estimate \rho. In Section 3.5 of the revised paper, we explained an estimation method of \rho and added some numerical results in Section 5.
>
>
> - Neither the language modeling task nor the OOD detection task make use of the upper bound.
>
> For the OOD detection task, numerical results with an adaptive choice of \rho were added in the revision.
>
> - the experimental section would benefit from including calibration metrics (ECE / Brier score; see 6).
>
> In language modeling tasks, we reported the perplexity that is the standard performance measure in this task. Though we can compute the calibration metrics such as ECE, the perplexity also provides a similar calibration criterion. The improvement of the perplexity indicates that our method yields the relevant calibration.

---

### Official Review · AnonReviewer3 · 2020-10-28
**Recommendation to Reject**

**Rating:** 3
**Confidence:** 3

**Review:**

This paper proposes a sampling free technique based on variance propagation to model predictive distributions of deep learning models. Estimating uncertainty of deep learning models is an important line of research for understanding the reliability of predictions and ensuring robustness to out-of-distribution data. Results are shown using synthetic data, perplexity analysis for a language modeling task and out-of-distribution detection performance using a convolutional network.

Overall I vote for rejecting the paper. The paper proposes an upper bound to the variance estimate of predictive distributions. However, the paper does not explain how the upper bound can be ensured. Furthermore, the experiments based on real data are conducted not using the upper bound approach but assuming strong independence assumptions (\rho = 0). In my opinion, the independence assumption needs extensive experiments to validate its performance under different scenarios. Furthermore, for further adoption of the upper bound approach, I think the authors need to provide extensive experiments to showcase the tradeoffs. For instance, as a reader of the paper I would like to get a sense of how much I will be overestimating the variance under typical scenarios. For this, I would recommend a formal analysis of the uncertainty estimates by inspecting the confidence intervals through coverage properties.

My specific comments related to this approach:
-	The authors propose two approaches in Section 3.1 for estimating the variance in predictions and the remaining subsections in Section 3 build on this.
o	The first approach (a) assumes independence in the input. In my opinion, this is a very strong assumption. For instance, multi-collinearity in features in DNN’s is a very common usage pattern. In domains like images, by construction of the problem, you expect a spatial correlation structure. Furthermore, the multi-layer perceptron architecture by construction is susceptible to correlation among variables. I would highly recommend adding a discussion on the validity independence assumption in general.
o	The second approach (b) will be necessarily true if `\rho >= \max_{j,j’:j!=j’} \rho_{j,j’}`. Only then this will guarantee that you will obtain an upper bound to the variance but it is very likely that you will overestimate the variance since your estimate is going to be bound by the highest correlation in your system. Again, in my opinion, this is a significant limitation of this approach and I recommend that authors highlight these points. Another minor thing to note in the manuscript is that when \rho=0, option (b) reduces to option (a).
o	I would suggest adding another summation term in Var(y_i) related equations to denote the double summation (indexed by j’) happening over the covariance terms (in Section 3.1).
o	Related to the above point, I could not follow the mathematical derivation from line (1) to line (2) in Equation (1). Could the authors provide an explicit derivation to ensure that the derivation is correct?
o	Furthermore, author’s claim that “distribution of y_i is well approximated by the univariate Gaussian distribution if the correlation among x is small” (Section 3.1) needs justification. To my knowledge, Wang & Manning (2013)’s Gaussian approximation holds due to central limit theorem as the number of samples approaches infinity but I do not think this will be applicable in this sample-free setting.

-	Regarding the results:
o	The results shown in Figure 1 demonstrates that choosing an appropriate `\rho can be challenging. We see underestimation of uncertainty with \rho <=0.15 and overestimation with \rho = 1. However, since these are based on synthetic data, I would suggest that the authors formally assess the fit using metrics like confidence interval widths and coverage.
o	The authors note that “Estimation of ρ is possible by observing the outputs of middle layers several times under the approximate predictive distribution.”. I am not convinced that this is an easy problem, I would recommend that the authors provide an example and elaborate this in depth.

Minor comments:
-	Please indicate the best results in the tables by highlighting them.
-	The uncertainty in deep learning literature typically employs distinction between aleatory and epistemic uncertainties. I think the manuscript can benefit how this proposed approach maps to the different sources of uncertainties.

---

> ### Author Response · Authors · 2020-11-21
> **To reviewer 3**
>
> Thank you very much for your constructive feedback and comments.
>
> - The first approach (a) assumes independence in the input. The second approach (b) will be necessarily true if \rho >= \max_{j,j’:j!=j’} \rho_{j,j’}.
>
> We think that the second approach is important to avoid the overconfidence for the prediction with the Bayesian method. So, we propose to use the second approach. Hence, we dropped the formula (a) and presented only (b) in the revision.
> In our method, the overestimation of the variance can occur. But, the relevant choice of rho provides a meaningful assessment of the uncertainty. In the numerical experiments of the revision, we added the method to select the relevant rho with a light computation cost.
>
> Also, we will supplement the detailed derivation of the equation in (b).
>
> - To my knowledge, Wang & Manning (2013)’s Gaussian approximation holds due to central limit theorem as the number of samples approaches infinity but I do not think this will be applicable in this sample-free setting.
>
> We think that the Gaussian approximation also holds in the sample-free setting. In Wang & Manning (2013) and other related papers such as Postels(2019), the Gaussian approximation is considered under also the condition that the dimension of input to the middle layer is sufficiently large.
>
>
> - I would suggest that the authors formally assess the fit using metrics like confidence interval widths and coverage.
>
> In the revision, we add the width of confidence intervals to the numerical results.
>
>
> - Estimation of rho
>
> There are several approaches to determine the relevant rho with a light computation. A simple method is to use the validation data to estimate \rho. In Section 3.5 of the revised paper, we explained an estimation method of \rho and added some numerical results in Section 5.

---

### Official Review · AnonReviewer4 · 2020-10-30

**Rating:** 4
**Confidence:** 4

**Review:**

Thank you for the interesting paper!

Summary

The authors focus on the important problem of efficient uncertainty quantification. More specifically, they propose a methodology that approximates the variance across samples from an MC dropout model with a single forward pass. To do this, they define analogs to existing layers such that they analytically (and often approximately) propagate variance from the input to the output of the layer. By repeating this for all layers, they construct a "variance-propagation BNN" (VPBNN) model that otherwise has the same architecture as the existing MC dropout model but is able to effectively output sample variance with a single forward pass. They demonstrate the effectiveness of their approach on (1) a simple synthetic problem, (2) a language modeling task, and (3) OOD detection.

Strengths

- Efficiently (from a FLOPS standpoint) propagating model uncertainty in a BNN is an important research area, particularly for compute-constrained use cases.
- The authors focus on a relevant set of experiments to demonstrate both the ability of their method to approximate MC dropout, and the ability to perform at, or better than, existing methods on downstream tasks.


Weaknesses

As noted below, I have concerns around the experimental results. More specifically, I feel that there is a relative lack of discussion around the (somewhat surprising) outperformance of baselines that VPBNN is aiming to approximate, and I feel that the experiments are missing what I see as key VPBNN results that otherwise leave the reader with questions. Additionally, I think the current paper would benefit from including measurements and discussion around the specifics of computational and memory costs of their method.

Recommendation

In general, I think this could be a great paper. However, given the above concerns, I'm currently inclined to suggest rejection of the paper in its current state. I would highly recommend that authors push further on the noted areas!

Additional comments

- p. 1: "The uncertainty is defined based on the posterior distribution." For more clarity it could be helpful to update this to say that the epistemic model uncertainty is represented in the prior distribution, and upon observing data, those beliefs can be updated in the form of a posterior distribution, which yields model uncertainty conditioned on observed data.
- p. 2: "The MC dropout requires a number of repeated feed-forward calculations with randomly sampled weight parameters in order to obtain the predictive distribution." This should be updated to indicate that in MC dropout, dropout is used (in an otherwise deterministic model) at test time with "a number of repeated feed-forward calculations" to effectively sample from the approximate posterior, but not directly via different weight samples (as in a variational BNN). With variational dropout, this ends up having a nice interpretation as a variational Bayes method, though no weight distributions are typically directly used with direct MC dropout.
- p. 2: Lakshminarayanan et al. (2017) presented random seed ensembles, not bootstrap ensembles (see p. 4 of their work for more info). They used the full dataset, and trained M ensemble members with different random seeds, rather than resampled data.
- p. 4: For variance propagation in a dropout layer with stochastic input, it's not exactly clear from the text how variance from the inputs and dropout is being combined into an output Gaussian. I believe using a Gaussian is an approximation, and while that would be fine, I think it would be informative to indicate that. The same issue comes up with local reparameterization for BNNs with parameter distributions, where they can be reparameterized exactly as output distributions (for, say, mean-field Gaussian weight dists) so long as the inputs are deterministic. Otherwise, the product of, say, two Gaussian RVs is non-Gaussian.
- p. 7: Figure 1 is too small.
- p. 7: "Estimation of ρ is possible by observing the outputs of middle layers several times under the approximate predictive distribution. The additional computation cost is still kept quite small compared to MC dropout." How exactly is $\rho$ estimated? Is it a one-time cost irregardless of data that can then be used for all predictions from the trained model? Without details, this seems like a key component that can yield arbitrary amounts of uncertainty.
- p. 7, 8: For the language modeling experiment, why do you think VPBNN was able to achieve lower perplexity values than MC dropout? The text generally focuses on VPBNN as an approximation to MC dropout, and yet it outperforms it. The text would greatly benefit from more discussion around this point.
- p. 8: For the OOD detection experiment, I'm surprised that $\rho = 0$ was the only VPBNN model used, since Section 5.1 and Figure 1 indicated that it led to overconfident models. Can you include results with other settings of $\rho$? Moreover, from Figure 1 we see that (for that model) VPBNN with $\rho = 0$ qualitatively yielded the same amount of predictive variance as the Taylor approximation. However, in Table 2, we see VPBNN with $\rho = 0$ outperform MC dropout (with 100 or 2000 samples) and the Taylor approximation. Why do you think this is the case, particularly if the standard deviation was used as the uncertainty signal for the OOD decision. I see that "This is because the approximation accuracy of the Taylor approximation is not necessarily high as shown in Section B", but I did not find Section B or Figure 3 to be clear. I think the text would benefit from more discussion here, and from the additional experiments for $\rho$.
- Can you include a discussion and measurements for FLOPS and memory usage for VPBNN? Specifically, given the discussion around efficiency and the implementation that doubles the dimensionality of the intermediates throughout the model, I believe it would be informative to have theoretical and possibly runtime measurements.

Minor

- p. 1: s/using the dropout/using dropout/
- p. 1: s/of the language modeling/of language modeling/
- p. 2: s/is the representative of/is representative of/
- p. 2: s/In the deep learning/In deep learning/
- p. 2: s/This relations/This relation/
- p. 5: Need to define $s$ as the sigmoid function in the LSTM cell equations.

---

> ### Author Response · Authors · 2020-11-21
> **To reviewer 4**
>
> Thank you very much for your constructive feedback and comments.
>
>
> - p. 1: "The uncertainty is defined based on the posterior distribution."
> - p. 2: "The MC dropout requires a number of repeated feed-forward calculations with randomly sampled weight parameters in order to obtain the predictive distribution."
> - p. 2: Lakshminarayanan et al. (2017)
> - Minor comments
>
> In the revision, we corrected the expressions.
>
> - p. 4: For variance propagation in a dropout layer with stochastic input, it's not exactly clear from the text how variance from the inputs and dropout is being combined into an output Gaussian.
>
> Since the stochastic input and the weight parameter in the dropout layer are independent, one can exactly calculate the variance of the product using each expectation and variance. We supplemented the details in the revision.
>
> - p. 7: "Estimation of ρ"
>
> There are several approaches to determine the relevant rho with a light computation. A simple method is to use the validation data to estimate \rho. In Section 3.5 of the revised paper, we added an estimation method of \rho and added some numerical results in Section 5.
>
> - p. 7, 8: For the language modeling experiment, why do you think VPBNN was able to achieve lower perplexity values than MC dropout?
>
> The VPBNN is not necessarily the approximation of MC dropout. Both MC dropout and VPBNN approximate the true posterior distribution, though MC dropout with a sufficient number of feed-forward calculations tends to provide a satisfactory result.
>
> - p. 8: For the OOD detection experiment, I'm surprised that rho=0 was the only VPBNN model used
>
> In the revision, we added the numerical result of VPBNN with an adaptive selection of rho. We find that a small rho is good to achieve high AUC values.
>
> - Can you include a discussion and measurements for FLOPS and memory usage for VPBNN?
>
> In the prediction phase, the computation time and memory cost of VPBNN is almost doubled compared to the standard feed-forward calculation since the mean value, and the variance should be computed. On the other hand, MC-dropout's computation time is linearly increased with the number of sampling. In Section 3.5, we show the computation cost of VPBNN with adaptive \rho is less than that of MC dropout. Also, in the experiments on OOD, we added a comment on the computation time.

---

### Author Response · Authors · 2020-11-21
**To all reviewers**

We thank all the reviewers for the constructive feedback and helpful comments. We have revised our paper by taking into account their suggestions. (During the rebuttal phase the page limit is 9 pages). The major changes are summarized below.

- We added Section 3.5 in which the estimation method of the correlation parameter, rho, is proposed.
- In numerical experiments, some numerical results using our method, VPBNN, with the adaptive correlation parameter were added.

---

### Decision · Program_Chairs · 2021-01-07
**Final Decision**

**Decision:**

Reject

**Comment:**

This paper proposes an approach to estimating uncertainty in deep neural network models that avoids the need to make multiple forward passes through a network or through multiple individual models in a posterior ensemble. In terms of strengths, this is an important and timely topic that is of significant interest. The paper is clearly written for the most part. In terms of weaknesses, the significance of the work is low. As the reviewers note, there are multiple questions around the experimental evaluation that remain unresolved following the author feedback and discussion. In particular, the authors do not compare to baseline MCMC methods like HMC/SGHMC that can yield gold standard estimates of posterior predictive uncertainty. While not feasible for large-scale models, MCMC methods provides crucial sanity checks for uncertainty estimation on small-scale (e.g., MNIST-scale) models. Posterior distillation methods like Bayesian Dark Knowledge are also not considered in the evaluation and should be compared to where the distillation computation is feasible. There are also foundational technical correctness issues with respect to uncertainty quantification due to the fact that the paper is approximating the measure of uncertainty produced by MC Dropout, which itself only approximates the true Bayesian posterior predictive distribution under additional assumptions. This makes empirical comparissons to MCMC methods all the more important. Following the discussion, the reviewers agree that the paper is not yet ready for publication.